



# Spatial Dependency in Nonstationary GEV Modelling of Extreme Precipitation over Great Britain

Han Wang[1], Yunqing Xuan[1]

[1]Zienkiewicz Centre for Computational Engineering, College of Engineering, Swansea University Bay Campus, Swansea SA1 8EN, UK.

*Correspondence to*: Yunqing Xuan (y.xuan@swansea.ac.uk)

**Abstract.** This paper presents a study on extreme precipitation using both stationary and non-stationary Generalized Extreme Value (GEV) models over a large number of samples distributed over Great Britain (GB) for the last century, aiming to gain insights in the spatial dependency of the GEV distribution. Not only L-Moments (LM) and Maximum Likelihood (ML) estimation methods but a Bayesian Markov-Chain Monte Carlo (B-MCMC) method are incorporated into the GEV models to characterize the uncertainty in the nonstationary risk-based assessment. The samples are generated using a toolbox of spatial random sampling for grid-based data analysis (SRS-GDA). The results show that a markedly large proportion (70%) of the samples are favour nonstationary assumption GEV models as far as the annual maximum daily rainfall (AMDR) is concerned. The most frequent AMDR, as represented by the location parameter tend to be increasing over the time for more than half of the samples and in contrast, only 8% have a downward trend. A spatially clustering pattern is also clearly present. For rarer (with 0.1 probability) AMDR, they are shown to have a tendency of becoming more extreme over time, for more than half of the samples. For the three methods, the LM method with stationary GEV maintain best results but for AMDR values with higher probability (5-year return level); the B-MCMC method with nonstationary GEV, however, outperform other combinations by a large margin for more extreme events (50-year return level). The findings suggest that an overhaul of the current engineering design storm practice may be needed in view of environmental change impact on natural processes.

## 1 Introduction

Extreme value (EV) theory and its application in modelling meteorological and environmental processes is a standard practice for designing and validating many infrastructure systems. Following this, a classic analysis approach is to use historical hydro-climatic data, such as rainfall, temperature, flood etc., to estimate the parameters of the required EV model which would offer probability distribution of the natural phenomenon in question, so as to address its occurrence or exceedance probability at given thresholds. Since Jenkinson (1955) proposed a generalized approach to analyzing the frequency distribution of annual maxima, many efforts have been put in quantifying the natural phenomena at extreme levels by using the Generalized Extreme Value (GEV) models (e.g. Gumbel, Fréchet, Weibull) with parameter estimation using the Maximum Likelihood method (ML)





and the L-Moments method (LM), especially in designing and planning water engineering systems (Coles and Tawn, 1996;
Lazoglou and Anagnostopoulou, 2017; Mannshardt-Shamseldin et al., 2010; Shukla et al., 2012; Yoon et al., 2015).

Recently, there has been a growing interest in explaining natural events from a climate-change perspective as many key hydro-climatic variables, e.g. precipitation, temperature, streamflow etc., are indeed changing due to the impact from climate change (Assani and Guerfi, 2017; Herring et al., 2018). In view of the reliability of infrastructure designs based upon extreme value analysis, stationary risk analyses have been re-assessed from a new adaptive perspective where Sarhadi et al. (2016) proposed
a multivariate time-varying risk framework for all stochastic multidimensional systems under the influence of the changing environment. For the commonly used GEV model, this is meant to assume its scale and location parameters to be varying with time or other climate indices. For example, Hasan et al. (2012) proposed two nonstationary GEV models for extreme temperature and each model assumes only one parameter as nonstationary depending linearly and exponentially on time respectively; Sarhadi and Soulis (2017) defined both the scale and location parameters for extreme precipitation analysis using
a linear, time-varying representation. Their results demonstrated underestimation of the extreme precipitation if stationary models are used instead; Panagoulia et al. (2014) generated 16 nonstationary GEV models for extreme precipitation with linear time dependence of location and log-linear time dependence of scale, employing the Akaike Information Criterion (AIC) and the Bayesian Information Criterion (BIC) for selecting the best model and examined confidence intervals for model parameters. Different from the researches listed above which assume a constant shape parameter, Ragulina and Reitan (2017) explored the
change of the shape parameter and found that it evidently depends on the elevation of study areas.

Although in the last few decades there have been several studies applying nonstationary GEV distribution to fit extreme rainfall, most of them focused on a limited number of specific domains because of data availability issues in hydrological observations; therefore, their conclusions are mostly of rationale and lack of generalization (Ganguli and Coulibaly, 2017). In addition, the performance of the existing GEV fitting methods, has not been systematically assessed as to their suitability,
especially in the context of fitting nonstationary models.

To address these issues, we present in this paper a study of extreme rainfall using both stationary and non-stationary GEV models over a large number of samples distributed over Great Britain (GB), aiming to gain insights in the spatial dependency of the GEV distribution as well as the performance of the existing three methods. There are 88 domain samples with the same size of 500 km$^2$, spatially distributed in the mainland of GB. These samples are generated by using the toolbox we developed
for spatial random sampling for grid-based data analysis (SRS-GDA) (Wang and Xuan, 2020). The underlying precipitation dataset is taken from the GEAR dataset (Gilbert, 1987) which covers the GB area with a spatial resolution of 1 km×1 km over the region of 700×1250 km$^2$. The quality and homogeneity of the GEAR dataset have been well tested by its provider, the Centre for Ecology & Hydrology (CEH) of the UK.

The main objectives of this study include: 1) to reveal the extreme rainfall pattern that varies with the time during the last
century in GB; 2) to assess the applicability of both stationary and nonstationary GEV models; 3) to test the three mainstream parameter estimation methods: LM, ML and a Bayesian Markov-Chain Monte-Carol (B-MCMC) with regards to their



goodness of fitness at different levels of rarity of rainfall extremes. The specific focus on the spatial dependency of the methods tested the highlights of this study.

The rest of this paper starts with presenting the main methodology used including parameter estimation for both stationary and nonstationary GEV models; it then follows the introduction of the study area and the datasets in Sect. 3; Sect. 4 shows the results alongside a detailed discussion focusing on the spatial feature of both stationary and nonstationary GEV models and their performances. The conclusions and recommendation are given in Sect. 5.

## 2 Methodology

We propose the following approach which covers the four related aspects of this study:

- Propose and fit the stationary generalized extreme value (S-GEV) model with fixed parameters to the time series obtained at every sampled domain;

- Propose and fit the nonstationary generalized extreme value (NS-GEV) model with time-varying parameters to the same time series with different parameter estimation methods applied;

- Evaluate the performance of the two types of models in various contexts with regards to the geographical locations, level of extremity as well the method of fit.

### 2.1 Stationary Generalized Extreme Value Model (S-GEV)

For a given sampled area, the annual maxima series of the areal daily rainfall is extracted and denoted as $X$. We then consider using the GEV to fit the series with the cumulative distribution function defined in Eq. (1) and its inversion (Eq. (2)) to obtain the threshold value $X_n$ at a different return level $F_n$.

$$F(x; \sigma, \mu, \xi) = \Pr(X \le x) = exp[-(1 + \xi(\tfrac{x-\mu}{\sigma}))^{-1/\xi}], \tag{1}$$

The cumulative probability function $F$ is defined for $\{1 + \xi(x - \mu)/\sigma > 0\}, -\infty < \mu < \infty, \sigma > 0$ and $-\infty < \xi < \infty$, where $\mu$ is the location parameter, $\sigma$ is the scale parameter, and $\xi$ is the shape parameter. There are three types of distributions from the GEV family which are distinguished by their shape parameters. The Type I GEV, also known as the Gumbel distribution, refers to the case where $\xi = 0$; while the type II and III are known as the Fréchet distribution and the Weibull distribution corresponding to the cases where $\xi > 0$ and $\xi < 0$ respectively.

The inverse form of the GEV distribution is given by Nascimento et al. (2016)

$$X_n = \begin{cases} \mu + \dfrac{\sigma}{\xi}[(-\ln F_n)^{-\xi} - 1)], & \xi \ne 0, \\ \mu + \sigma \times \ln[-\ln F_n], & \xi = 0, \end{cases} \tag{2}$$

Equation 1 and Equation 2 represent the stationary GEV (S-GEV) model whose parameters are independent and invariable with time or other covariations, hence the name. The parameters of the S-GEV model are estimated by using the L-Moment





(LM) method (Hosking, 1990; Hosking and Wallis, 2005) which is a common choice. The linear moments are the expectations of certain linear combinations of order statistics which contain the estimated parameters. For the GEV distribution ($\xi \neq 0$),
the first three linear moments are:

$$L_1 = \mu + \frac{\sigma}{\xi}[1 - \Gamma(1 + \xi)] = \beta_0, \tag{3a}$$

$$L_2 = \frac{\sigma}{\xi}(1 - 2^{-k})\Gamma(1 + \xi) = 2\beta_1 - \beta_0, \tag{3b}$$

$$\frac{L_3}{L_2} = \frac{2(1 - 3^{-\xi})}{(1 - 2^{-\xi})} - 3 = \frac{6\beta_2 - 6\beta_1 + \beta_0}{2\beta_1 - \beta_0}, \tag{3c}$$

where $\beta_r$ ($r = 0,1,2$) indicates the expectations of the quantiles or non-exceedance probabilities of the $r$-th random variable $X$ (i.e. the $r$-th AMDR) if the expectation $E[X]$ exists. They can be calculated by using the probability-weighted moment estimator which is given by

$$\beta_r = E\{X[F(X)]^r\}, \tag{4}$$

After the three linear moments are estimated, an approximate explicit solution for the shape parameter $\xi$ in the interval $-0.5 \leq$
$\xi \leq 0.5$, is calculated by using Eq. (5) (Hosking et al., 1985).

$$\xi = 7.8590\left(\frac{2L_2}{L_3 + 3L_2} - \frac{\ln 2}{\ln 3}\right) + 2.9554\left(\frac{2L_2}{L_3 + 3L_2} - \frac{\ln 2}{\ln 3}\right)^2, \tag{5}$$

The other two parameters can then be estimated by plugging back $\xi$ into Eq. (3).

## 2.2 Nonstationary Generalized Extreme Value Model (NS-GEV)

Compared with the S-GEV model, the nonstationary GEV (NS-GEV) model makes an important extension by assuming that the parameters change over elapsing time. In this study, the scale and location parameters are considered to vary monotonically
and linearly with time (see Eq. (7)) and thus its cumulative probability function and inversion are given as:

$$F_t(x; \sigma_t, \mu_t, \xi) = exp[-(1 + \xi(\frac{x - \mu_t}{\sigma_t}))^{-1/\xi}], \tag{6a}$$

$$X_n = \begin{cases} \mu_t + \frac{\sigma_t}{\xi}[(-\ln(Pr(X \leq x)))^{-\xi} - 1)], & \xi \neq 0, \\ \mu_t + \sigma_t \times \ln[-\ln(Pr(X \leq x))], & \xi = 0, \end{cases} \tag{6b}$$

Basically, the CDF $F_t$ follows the same form as the stationary one with an additional subscript $t$ added to the location and scale parameters which indicates that both parameters are time dependent. The shape parameter, $\xi$ is assumed to be constant. The linearly time-varying parameters are further shown in Eq. (7).

$$\begin{cases} \sigma_t = \sigma_0 + \sigma1 \times t, \\ \mu_t = \mu_0 + \mu1 \times t, \end{cases} \tag{7}$$

The NS-GEV model thus has five parameters $\{\sigma_0, \sigma1, \mu_0, \mu1, \xi\}$ which are denoted by a vector form $\boldsymbol{\theta}$ to help our discussion.
The LM method which is previously applied to estimate the parameters of the S-GEV model is unsuitable for the case of NS-





GEV; therefore, in this study, the Maximum Likelihood (ML) method is employed to estimate parameters and the Bayesian Markov-Chain Monte-Carlo (B-MCMC) method is incorporated into NS-GEV model to characterize the uncertainty.

- The ML method

The ML method (Myung, 2003) is built upon the likelihood function of the occurrence of the annual maximum daily rainfall
(AMDR) $x_t$:

$$L(x_t; \boldsymbol{\theta}) = \prod_{t=1}^{n} f(x_t; \boldsymbol{\theta}), \tag{8}$$

where $f(\cdot)$ is the univariate density function and $n$ is the length of dataset $x$. Its product is the likelihood function $L$. The set of the parameter $\boldsymbol{\theta}$ can then be obtained by maximizing the likelihood function:

$$\frac{\partial L(x_t; \boldsymbol{\theta})}{\partial \boldsymbol{\theta}} = 0. \tag{9}$$

Often, Eq. (9) cannot be solved analytically and in this study a numerical scheme was applied to obtain the three parameters.

- The B-MCMC method

The B-MCMC method makes use of the Bayesian inference to estimate the posterior distribution of the time-varying location and scale parameters $\boldsymbol{\theta}$ of the NS-GEV model. In this study, the estimated parameters of the S-GEV model are used to define the initial prior values of the NS-GEV model. The prior distribution of parameters is assumed to be a uniform distribution. Equation 10 presents the transformation from prior distribution to posterior distribution by multiplying by its likelihood (Rasmussen and Ghahramani, 2003).

$$p(\boldsymbol{\theta}|x, t) \propto p(x|\boldsymbol{\theta}, t) \times p(\boldsymbol{\theta}|t) = \prod_{t=1}^{n=113} p(x_t|\boldsymbol{\theta}_t, t) \times p(\boldsymbol{\theta}|t), \tag{10}$$

Where $p(x|\theta, t) \propto L(x; \theta, t)$ is the likelihood function and $p(\theta|t)$ is theprior probability distribution of the parameters $\boldsymbol{\theta}$; $t$ indicates the state.

Numerical iterations for processing the posterior distribution are carried out by using MCMC simulation (Binder et al., 2012; Manly, 2018; Metropolis and Ulam, 1949), which is also aimed at analyzing the uncertainty of the NS-GEV model. The final simulation results are compared with those estimated using the ML method.

The inputs to the B-MCMC method include: the initial values of the parameters taken from the S-GEV model, the likelihood function, the prior probability and the step set-up function which returns the step length of each iteration. For this study, a random step length is used. The length of the Markov chain is set as 15,000 which is long enough for the simulation; a value 1 is used for setting the skip set-up (N) to thin the chain by only storing every N steps.

Suppose that $S_t$ is the current (known) state with a prior probability $p(\boldsymbol{\theta}|t)$ and $S_{t+1}$ is the next-step state (unknown) with an
a prior probability of $p(\boldsymbol{\theta}'|t)$; an MCMC iteration can be described by the following steps and the flowchart shown in Fig. 2 (Carlo, 2004):

1) Propose a new step state $S_{t+1}$ by following a random walk and calculate the prior probability of $p(\boldsymbol{\theta}'|t)$ of this state; in the meantime, drawing a random number $p^*$ from $U(0,1)$.





2) If $\min\left(1, \frac{p(\boldsymbol{\theta}'|t)}{p(\boldsymbol{\theta}|t)}\right) \geq p^*$, then calculate the likelihood $p(x|\boldsymbol{\theta}', t)$ of $S_{t+1}$ and go to step 3, otherwise reject this state

and go back to step 1 to regenerate a state;

3) If $\min\left(1, \frac{p(x|\boldsymbol{\theta}', t)}{p(x|\boldsymbol{\theta}, t)}\right) \geq p^*$, then accept $S_{t+1}$ and store its parameters and go to the step 4, otherwise reject this state

and go back to step 1;

4) Check the iteration with the length of Markov chain, if the number of iterations is less than 15000, continue executing the loop (step 1 to 4); otherwise, finish the Monte-Carlo simulation and analyze the estimated parameters.

Finally, a quantile-quantile plot (Q-Q plot) is produced to compare the quantiles simulated by both the S-GEV and NS-GEV models against the empirical quantiles. The Q-Q plot has a reference line along which the data indicates the equalization between simulations and observations. The larger the deviation from this reference, the worse the performance of the model (S-GEV or NS-GEV) or method (LM, ML or B-MCMC).

2.3 Goodness of Fitness and Performance of S-GEV and NS-GEV Models

The Kolmogorov-Smirnov (K-S) Goodness of Fitness test (Kolmogorov, 1933; Smirnov, 1948) is widely used to assess the quality of the convergence of GEV distribution with the extreme hydro-climatic datasets. The test is carried out by comparing the empirical cumulative probability distribution with the GEV cumulative probability distribution. The maximum difference between the two distributions is used to covert the $p$-value which indicates whether the testing dataset follows the assumed distribution. The null hypothesis in this study is that the data are drawn from GEV distribution. The K-S test rejects the null

hypothesis if the $p$-value is below the significance level of 5% in this study.

Meanwhile, the difference between empirical designed rainfall ($y$) by its empirical cumulative probability distribution at different return periods and its counterparts ($y'$) by S-GEV and NS-GEV models, is applied to show the performance of GEV models and uncertainty arose from stationary and nonstationary assumptions, as defined in Eq. (11).

$Diff = y' - y,$            (11)

Small absolute value of *Diff* can be related to a less uncertainty and a better performance. In order to show such variation, a

boxplot is employed to indicate the under/overestimate the risk of extremes.

**3 Dataset and Study Area**

The dataset used in this study, named GEAR, is a gridded daily rainfall at a spatial resolution of 1 km × 1 km from 1898 to 2010 over Great Britain (GB) (Tanguy et al., 2016). The rainfall estimates are derived from the UK Met Office national database of observed precipitations. To derive the estimates, the precipitations from the UK rain gauge network were used.

The Natural Neighbor interpolation method, with an extra normalization step based on average annual rainfall, was used to generate the daily estimates. The estimated rainfall on any given day refers to the rainfall amount precipitated in the 24 hours between 9am on the day of report until 9am on the following day. The origin of the GEAR data matrix starts from the location





400 km west, 100 km north of the true Origin (49°N, 2°W), spreading 700 km east-westly and 1250 km south – northly. Figure 1b shows the GEAR data matrix where only the grids within the green area (over the mainland) were used in this study.

The SRS-GDA toolbox (Wang and Xuan, 2020)is then employed to generate samples of areas over the study domain. This toolbox can generate samples with either randomly or manually defined properties, e.g., location, size, shape and total number of samples, from the entire study area. In this study, each of the samples is predefined with a fixed size of 500 km$^2$ and a fixed spatial property $sp = 0.8$. The $sp$ is an important parameter that indicates the irregularity of the shape of areas sampled, expressed as the ratio of the north/south dimension of the domain in question over the east/west dimension. The value 0.8 was

used to guarantee a regular shape of the domains generated. It should be noted that the SRS-GDA toolbox is able to randomize location, size as well as shapes; in this study, however, our focus is on the impact of location only. One of such samples is shown in Fig. 1a which consists of 500 grids with a grid size of 1km × 1km, e.g. the same resolution as that of the GEAR dataset. The sampling is then repeated with randomized (non-overlapping) locations with a spatial interval of 40 km until finally we obtained 88 such samples located all over the study domain (see Fig. 1b).

## 175    4 Results and Discussion

### 4.1 Simulation Results of the S-GEV and NS-GEV Models

The parameters of the GEV distribution under both the stationary and nonstationary assumptions, are estimated by using the three methods (LM, ML and B-MCMC) for the 88 samples. The $p$-values, which indicates the goodness of fitness, are all very close to 1.0, which indicate a failure on rejecting the null hypothesis, i.e., the AMDR follows the GEV distribution at 5%

significance level. It should be noted that the high $p$-values cannot be used to confirm that the AMDR follows the GEV distribution, however, we follow other researchers here to use them to indicate that the AMDR is highly likely to follow the GEV distribution (De Michele and Avanzi, 2018; Hasan et al., 2012; Machiwal and Jha, 2008; Martin, 2013). Meanwhile, the *Diff* is applied to identify the best performance and uncertainty on nonstationary-based assumption.

Figure 3a shows the spatial distribution of the best selected GEV model for each sample. About 30% of the samples (27 over

88) show that the S-GEV model works better than the NS-GEV model under the linear time-dependent parameter assumption. Among those 70% samples favoring the nonstationary assumption, the B-MCMC method always converge to  better results than the ML method does. Geographically, the samples that favor stationary models (labelled by crosses) concentrate around the region of 100 km north in the vicinity of Manchester and Liverpool, with several others distribute in Southern England.

Figure 3b presents the spatial distribution of the best selected GEV models in terms of their types. Out of all samples, there

are more than half (55.7%) following the Fréchet distribution ($\xi > 0$), mainly located in Southern England, 37.5% following the Weibull distribution ($\xi < 0$) and the rest following Gumbel distribution ($\xi = 0$).

Figure 3c shows the spatial distribution of the samples with regards to how the parameters of the GEV distributions vary with time, i.e. the two parameters $\sigma 1$ and $1$ . The results are further summarised in Table 1 and Table 2 with more discussions in the following section.



## 4.2 Spatial Nonstationary Patterns of AMDR in GB

It is worth revisiting the implication of time-varying scale and location parameters of the GEV models. The scale and location parameters determine the shape and location of the GEV distribution (Kantar and Şenoğlu, 2008; Mann, 1967). The location parameter $\mu$ indicates the mode of the time series, which in our case, is related to the AMDR that has the most frequent occurrence. An increasing $\mu$ means that the AMDR values of the highest probability goes upward. The scale parameter $\sigma$ is related to the deviation of the AMDR values from $\mu$, which determines the stretch (for increasing $\sigma$) or squeeze (for decreasing $\sigma$) of the GEV probability distribution curve. The larger the scale parameter, the more spread-out the distribution is. Conversely, the smaller the parameter, the more compressed the distribution is. In our study, if $\sigma$ is estimated to be increasing with the time, the occurrence probability of extreme AMDR, i.e. rainfall ranked in the higher positions is increased.

As seen in Fig. 3c, several intriguing yet remarkable patterns can be found with regards to the fitted sale and location parameters:

- Most of the samples are in favor of the NS-GEV model, with only 30% samples are shown to have stationary $\mu$ and $\sigma$.

Geographically, these 30% samples are centered around 100 km north (the vicinity of Manchester and Liverpool) with only a few distributed in southern England and Scotland. One of such samples is examined to reveal the difference among the models and the combination of the three methods, as seen in Fig. 4. This sample is located to the west of Glasgow with a location index of (240 km, 660 km).

Figure 4 presents the observed AMDR over the entire period of 113 years, comparing the simulated series fitted by the two models (S-GEV and NS-GEV) using the three different methods discussed above (LM, ML and B-MCMC). The majority of the AMDR values, which are related to $\mu$ and can be regarded as the most frequent rainfall, fluctuate between 40mm and 60mm during the entire period. And such fluctuations, which are related to $\sigma$, are even and have no perceivable changes from the first to the second 50 years.

- About 56% samples are detected to have an increasing $\mu$

As mentioned previously, an increasing location parameter indicates an upward trend of the most frequent AMDR values. It is found that more than half of the samples over GB demonstrate such increasing trend. In fact, if including those samples with $\mu = 0$, there are 92% of samples coming with non-decreasing $\mu$. Location wise, samples with increasing $\mu$ generally are from the middle England and Wales, the Lake District and the Highlands. Figure 5 presents three examples with their characteristics shown in Table 1. It is also worth noting that more than half of these samples come with an increasing scale parameter $\sigma$ while less than a quarter of them have a decreasing one. This implies that not only are the AMDR in majority samples getting higher on average, they also are becoming more extreme in those areas. It is also clear from Table 1 that the changing scale parameter $\sigma$ with an increased $\mu$ in the first two example samples, which represents 70% of all samples, leads to an increasingly more frequent 1-in-50-year rainfall over time; only the third sample, representing the rest 30%, whose dropping $\sigma$ makes such rainfall rarer as an 1-in-60-year event. This corroborates, quantitively, with other studies suggesting that extreme rainfalls are



more likely to have become more frequent, or in other words, the return level of the events that engineering designs rely on could possibly be reduced and become less reliable.

- Only 8% samples present a decreasing trend of most frequent AMDR values.

Differing from the first two patterns, there are only 7 over 88 samples showing a decreasing trend in their $\mu$ parameters. Remarkably, these samples are all located in Scotland. Figure 6 presents three examples with their characteristics shown in Table 2. Except the first sample, the most samples with a decreased $\mu$ and unchanged or decreased $\sigma$ show 1-in-50-years rainfall become rarer after 100 years, especially when $\sigma$ drops significantly.

## 4.3 Performance of Methods

In order to compare the three statistical methods LM, ML and B-MCMC, we divide the AMDR values by their associated probability P into four levels separated by: $P_{50}$, the 50[th] quantile (or 1-in-2 years in terms of return level); $P_{80}$, the 80[th] quantile (1-in-5 years); $P_{98}$, the 98[th] (1-in-50 years) and even $P_{99}$, the 99[th] (1-in-100 years) quantiles of their empirical CDF's:

- L1: $P \leq P_{50}$;
- L2: $P_{50} < P \leq P_{80}$;
- L3: $P_{80} < P \leq P_{98}$;
- L4: $P > P_{98}$;

These four levels can be considered as the low (L1), the medium (L2), the high (L3) and the very high AMDR (L4). The higher the AMDR is, the less frequently it appears. Therefore, L4 is considered to be the extreme case. The assessment of the three fitting methods are then conducted on a level-by-level basis over the entire 113 years for all sampled domains.

Q-Q plot is used to compare the performance of the three methods with one example shown in Fig. 7a. The reference line (dash line) indicates the perfect fit. Larger deviation from this reference line implies worse performance of the fitted GEV model. It is interesting to see for this sample, the LM method (for S-GEV) and the ML (for NS-GEV) both work better for the lower quantile of rainfall (below L1) where the B-MCMC method tends to a bit underestimation. For the medium quantiles, e.g. L2 and L3, all three methods achieve similar level of performance. It is the extreme case, e.g. L4 that the B-MCMC method gets 250 a clear leading edge with much closer results. This pattern of performance not only appears for the selected sample, but it also represents most of all samples when plotting the simulation *Diff*s (Eq.(11)), as shown in Fig. 7b. Again, for the quartile rainfall less than 10years return level (i.e., L1 and L2), the smallest difference and least uncertainty are observed in the model results by S-GEV with the LM method; for L4, the boxplot of the difference by S-GEV is skewed left with 2 outliers while NS-GEV by B-MCMC method show a much smaller uncertainty without outliers and less difference but a bit underestimation.

Furthermore, the uncertainty grows as the return period increases. More details can be found in Table 3.

The spatial distribution of the best selected methods with regards to 4 levels are further summarized in Fig. 8 and Table 4. They confirm the said pattern change, i.e., the LM methods dominates the L1 level and gradually, the ML method gets more





and more contribution as the best performers from L2 to L3. And again, for the extreme case (L4), the B-MCMC is a clear winner.

It is also interesting to interpret Table 4 from another perspective of model choice. For low and medium AMDR, the stationary model, e.g. S-GEV can sufficiently represent them very well; however, for higher level, NS-GEV models are preferred. This also implies that there have been more possible time-varying changes to the higher level of AMDR than to the normal, medium range of AMDR.

## 5 Conclusions

We present a study of spatial dependency of both stationary and nonstationary GEV modelling of annual maximum daily rainfall over Great Britain for the period of 113 years using a large grid-based dataset GEAR. We also demonstrate the performance of the three most commonly used fitting methods LM, ML, B-MCMC, particularly over different level of rarity of the event. The study is assisted with a toolbox of spatial random sampling (SRS-GDA) which provides enough samples. We find that:

1) In general, 70% samples favor the NS-GEV model whose parameters $\mu$ and $\sigma$ are assumed to be linearly changing with time;

2) Among those NS-GEV applications, B-MCMC always performs better than ML. However, S-GEV model estimated by the LM method is the best chose for modelling the rainfall less than 1-in-5-year return level while NS-GEV model incorporated by the B-MCMC method prevails for the extreme cases, e.g. the rainfall higher than 1-in-50-year return level;

3) More than half of those samples favoring the NS-GEV model show a continuous increase on $\mu$ which is related to the increase of the most frequent AMDR in those samples. Meanwhile more than half of them are further accompanied by an increase on $\sigma$, which leads to an overall dropping of the return level from 1-in-50-year to 1-in-20 year over the study period of 113 years. If translated into everyday language, this means that not only do the most frequent events (w.r.t. $\mu$) becomemore extreme, the extreme events also become more frequent (w.r.t. $\sigma$).

We trust that the findings from this study are of great relevance as they not only further corroborate other research findings on extreme rainfall, e.g. extreme events are likely to become more frequent due to climate change impact, but they also quantitatively address how such changes may affect the original engineering design standard. The fact that the combination of NS-GEV/B-MCMC always perform the best for evaluating the extreme events regardless of the GEV model, may inspire a reconsideration of the current practice of designing storms.

Further work is recommended to have a closer look at the underlying datasets with respect to the potential inconsistency in the resolution of the data observed near the west coast of Scotland. In addition, comparative study with long-term, single gauge observations, as well as catchment orientated sampling will make conclusions more robust.



**Code/Data availability**

The GEAR dataset for this study is provided by Centre of Hydrology and Ecology (CEH). The open-source toolbox of spatial
random sampling for grid-based data analysis (SRS-GDA) developed by authors to generate samples used in this study can be
obtained from https://github.com/wanghan924/SRS-GDA_Toolbox.git.

**Author contribution**

There is a contribution table where Y indicates the author contributed to this activity.

| Contribution activities | Han Wang | Yunqing Xuan |
|---|---|---|
| Conceptualization | | Y |
| Data curation | Y | |
| Formal analysis | Y | |
| Funding acquisition | | Y |
| Methodology | Y | Y |
| Project administration | | Y |
| Software | Y | |
| Supervision | | Y |
| Visualization | Y | |
| Writing – original draft | Y | |
| Writing – review & editing | Y | Y |

**Competing interests**

The authors declare no competing interests.

**Acknowledgments**

The authors would like to thank the Centre of Hydrology and Ecology (CEH) for providing the GEAR dataset. The open-
source toolbox of spatial random sampling for grid-based data analysis (SRS-GDA) developed by authors to generate samples
used in this study can be obtained from https://github.com/wanghan924/SRS-GDA_Toolbox.git. This research is supported by
the Chinese Scholarship Council, China and the College of Engineering, Swansea University, UK via their PhD scholarships
offered to the co-author Han Wang and the Royal Academy of Engineering UK-China Urban Flooding Programme Grant
(REF: UUFRIP\10021), which are both gratefully acknowledged.



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





**Table 1. Example of three types of samples all with an estimated increasing $\mu$, but with increasing, unchanged and decreasing $\sigma$ respectively**

| | Figure 6a | Figure 6b | Figure 6c |
|---|---|---|---|
| Location (x,y) | 360km, 660km | 400km, 340km | 280km, 220km |
| Location description | Around 60 km south-east of Edinburgh | Around 40 km west of Nottingham | Around 60 km north-west of Cardiff |
| $\mu$ | Increasing circa 6mm/100 yr | Increasing circa 4 mm/ 100yr | Increasing circa 5 mm/100 yr |
| $\sigma$ | +2 /100 yr | Unchanged | -2/100 yr |
| Most frequent AMDR | First 50 years: around 30mm; Last 50 years: around 40mm. | First 50 years: around 30mm; Last 50 years: around 40mm. | First 50 years: around 30mm; Last 50 years: around 40mm. |
| Change of a ref return level 1-in-50 yr | 1-in-16-yr | 1-in-26-yr | 1-in-60-yr |
| Number (percentage) of the same type samples | 26 (53%) | 11 (22%) | 12 (25%) |

**Table 2. Example of the three types of samples all with an estimated decreasing $\mu$, but with increasing, unchanged and decreasing $\sigma$ respectively.**

| | Figure 7a | Figure 7b | Figure 7c |
|---|---|---|---|
| Location (x,y) | 400km, 620km | 280km, 620km | 280km, 740km |
| Location description | Around 90 km south-west of Edinburgh near the coast | Around 60 km south-west of Glasgow | Around 90 km northeast of Glasgow |
| $\mu$ | Decreasing circa 1 mm/100 yr | Decreasing circa 3 mm/100 yr | Decreasing circa 1 mm/100 yr |
| $\sigma$ | Increase 2/100 year | unchanged | Decrease -3/100 year |
| Most frequent AMDR | First 50 years: around 45mm; Last 50 years: around 42mm. | First 50 years: around 50mm; Last 50 years: around 40mm. | First 50 years: around 30mm; Last 50 years: around 40mm. |
| Change of a ref return level 1-in-50 yr | 1-in-27-yr | 1-in-68-yr | 1-in-83-yr |
| Number (percentage of the same type samples | 2 (29%) | 3 (57%) | 2 (29%) |






**Table 3. Performances of stationary and nonstationary GEV models by three methods in reproducing the quantile associated to the empirical cumulative frequency for the 88 samples in the past 113 years corresponding to boxplot.**

| Return period | Methods | Lower whisker (mm) | 1st quartile (mm) | Median (mm) | 3rd quartile (mm) | Upper whisker (mm) | IQR (mm) | Number of outliers |
|---|---|---|---|---|---|---|---|---|
| 2 years | LM | -1.1 | -0.3 | 0.1 | 0.7 | 1.8 | 1.0 | 0 |
| | ML | -1.5 | -0.2 | 0.3 | 0.7 | 1.3 | 0.9 | 2 |
| | B-MCMC | -1.8 | -0.5 | -0.1 | 0.5 | 1.8 | 1 | 3 |
| 5 years | LM | -2.0 | -0.7 | 0 | 0.5 | 1.9 | 1.2 | 4 |
| | ML | -2.8 | -0.7 | 0.2 | 1.1 | 3.1 | 1.8 | 2 |
| | B-MCMC | -4.4 | -1.7 | -0.5 | 0.3 | 2.9 | 2.0 | 0 |
| 10 years | LM | -4.4 | -1.1 | 0.5 | 1.6 | 4.1 | 2.7 | 3 |
| | ML | -4.2 | -0.5 | 0.7 | 2.1 | 4.5 | 2.6 | 2 |
| | B-MCMC | -5.4 | -2.2 | -0.9 | 0.3 | 3.9 | 2.5 | 4 |
| 50 years | LM | -10.6 | -1.6 | 2.0 | 4.5 | 12.2 | 6.1 | 4 |
| | ML | -6.8 | 0.4 | 3.6 | 6.2 | 13.7 | 6.6 | 1 |
| | B-MCMC | -9.8 | -5.1 | -1.9 | 0.5 | 4.5 | 5.6 | 1 |
| 100 years | LM | -15.6 | -1.3 | 4.3 | 8.6 | 20.5 | 9.9 | 2 |
| | ML | -10.7 | 1.5 | 5.4 | 10.4 | 21.8 | 11.9 | 3 |
| | B-MCMC | -16.2 | -5.3 | -1.2 | 2.5 | 12.3 | 7.8 | 0 |

**Table 4. The number of best selected methods for simulating AMDR's with respect to the four levels.**

| AMDR levels | Low | Medium | High | Very high |
|---|---|---|---|---|
| LM | 67 | 69 | 41 | 6 |
| ML | 19 | 8 | 19 | 3 |
| B-MCMC | 2 | 11 | 28 | 79 |




**Figure 1. The sampling area as shown in (a) the base data grids of a single sample; (b) the spatial distribution of the 88 samples.**

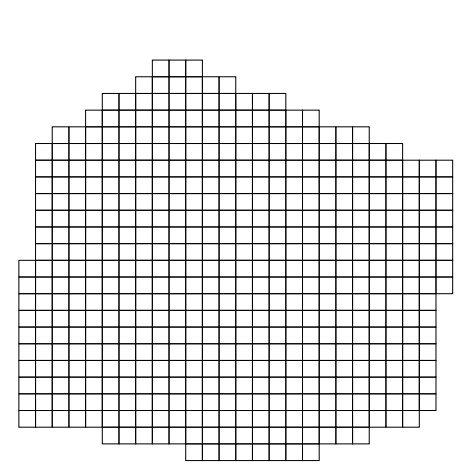

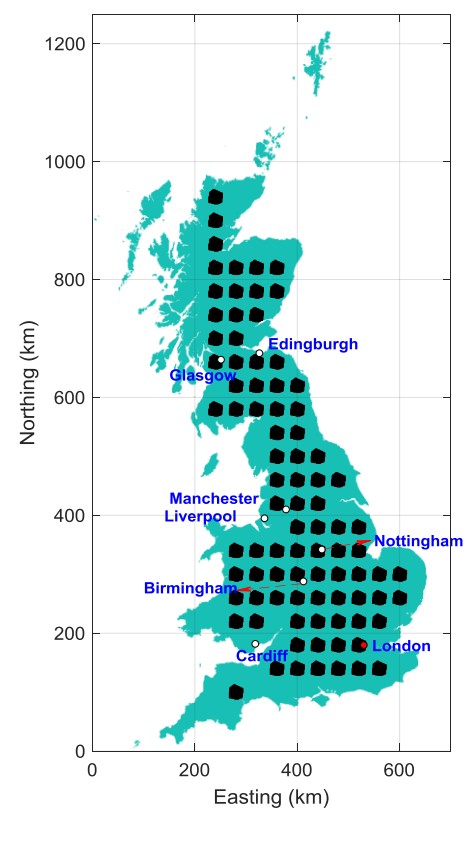

(a)                                        (b)





**Figure 2. The process of B-MCMC simulation.**

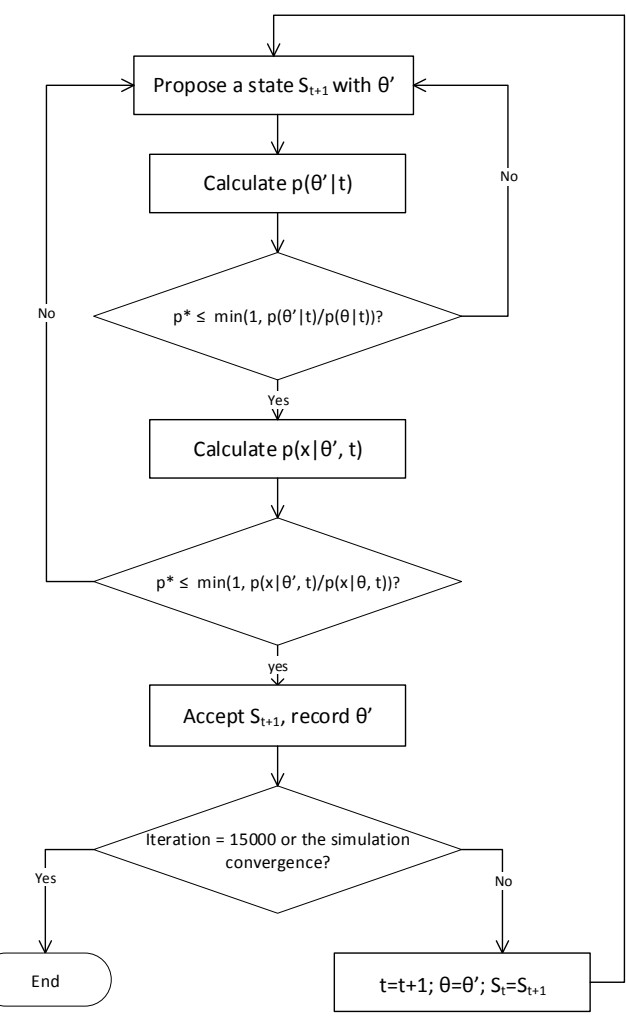







**Figure 3. (a) Spatial distribution of the best fitted models (S-GEV or NS-GEV); (b) Spatial distribution of the GEV type of the best fitted models (Gumbel, Fréchet or Weibull); (c) Spatial distribution of the changing scale and location parameters; (d) Summary of**
**the changing scale and location parameters.**







**Figure 4. Observed and simulated AMDR of the sample located at 240km, 660km with stationary $\mu$ and $\sigma$.**

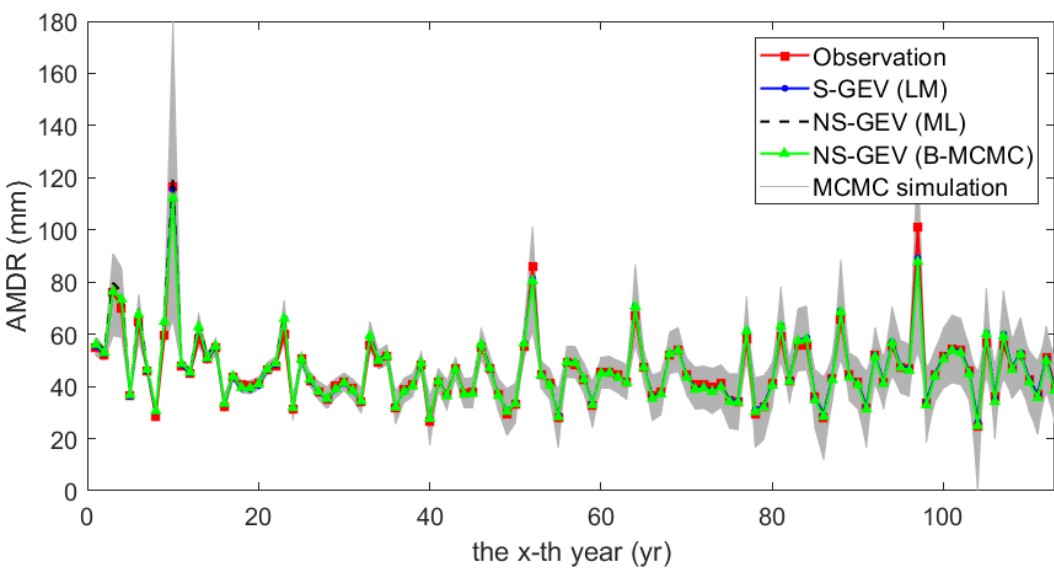


**Figure 5. Observed and simulated AMDR's of the samples located at (a) 360km, 660km with both increasing $\mu$ and $\sigma$.; (b) 400km, 340km with an increasing $\mu$ and an unchanged $\sigma$; (c) 280km, 220km with an increasing $\mu$ and a decreasing $\sigma$.**

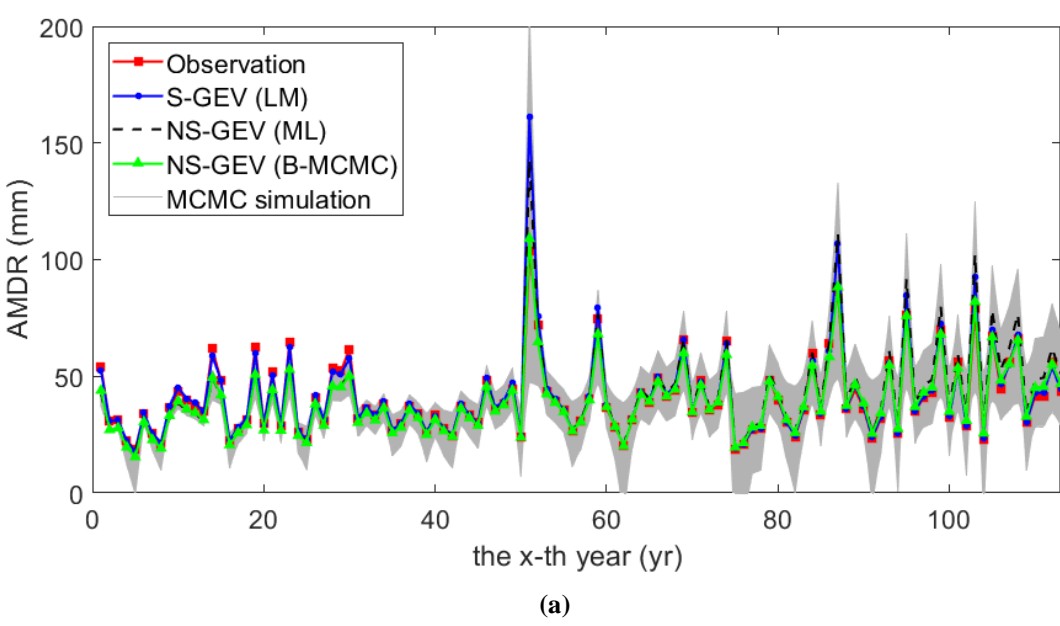

**(a)**





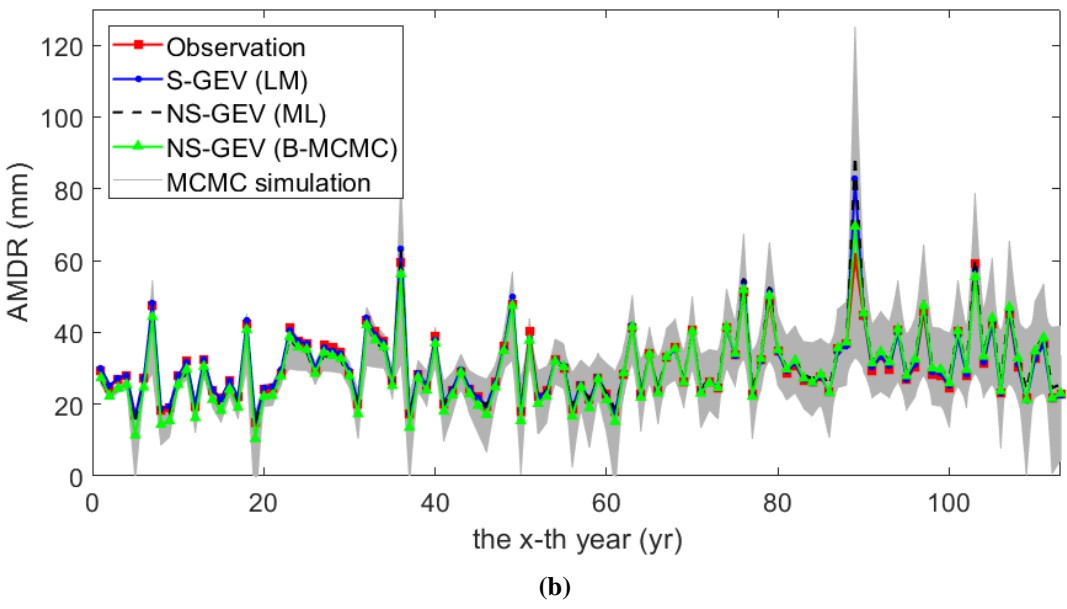

**(b)**

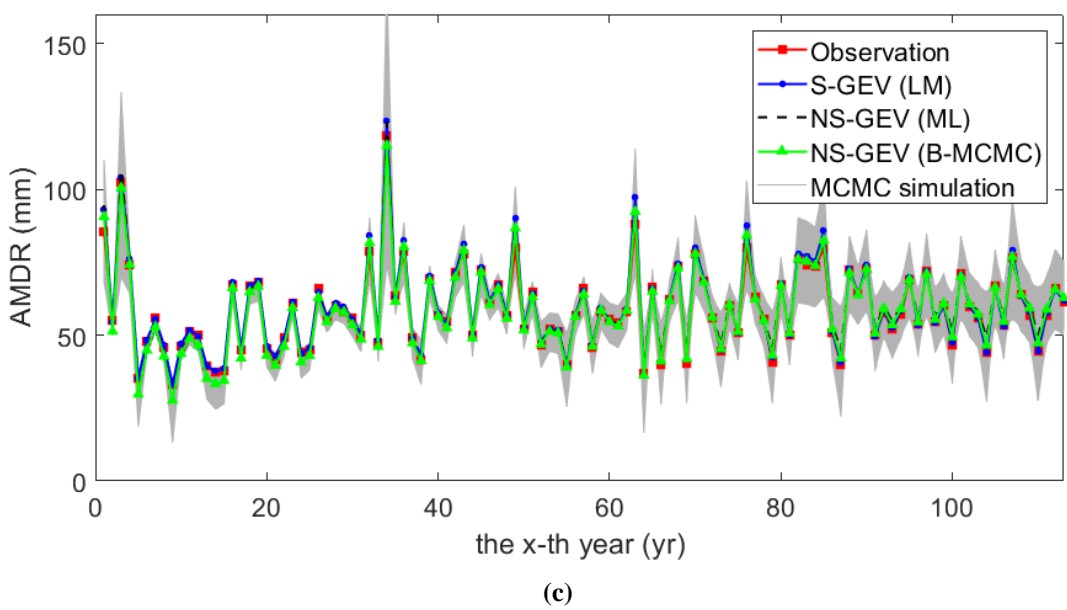

**(c)**






**Figure 6. Observed and simulated AMDR's of the samples located at(a) 400km, 620km with a decreasing $\mu$ and an increasing $\sigma$.; (b) 280km, 620km with a decreasing $\mu$ and an unchanged $\sigma$; (c) 280km, 510km with both decreasing $\mu$ and $\sigma$.**

**(a)**

**(b)**



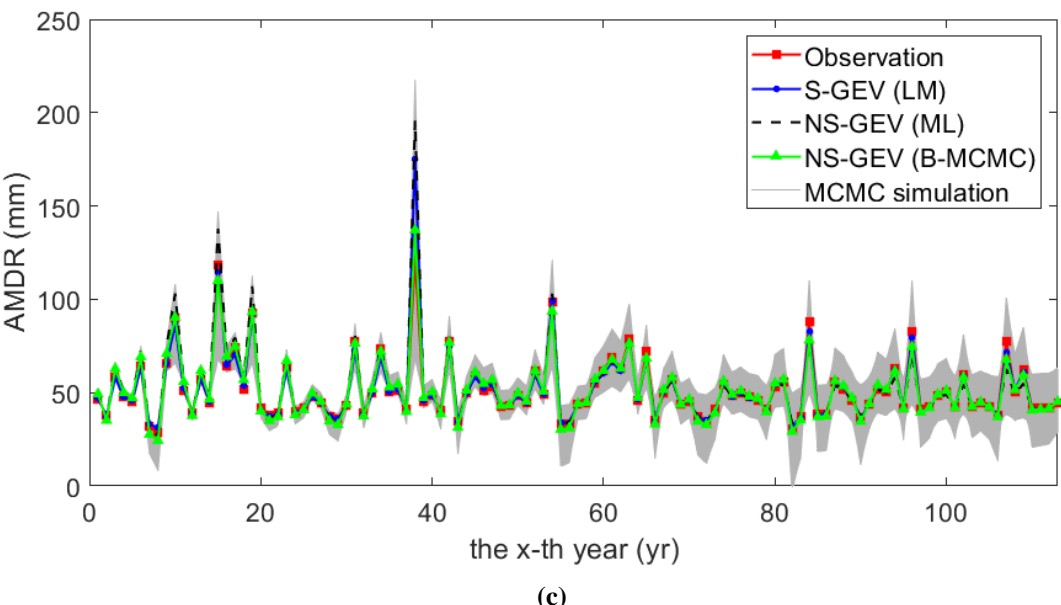

**(c)**

**Figure 7. (a) Q-Q plot of the simulated AMDR's of an example sample with the location index of (320km, 660km); (b) the difference**
**between GEV-modelled and empirical daily extremes at different return periods.**

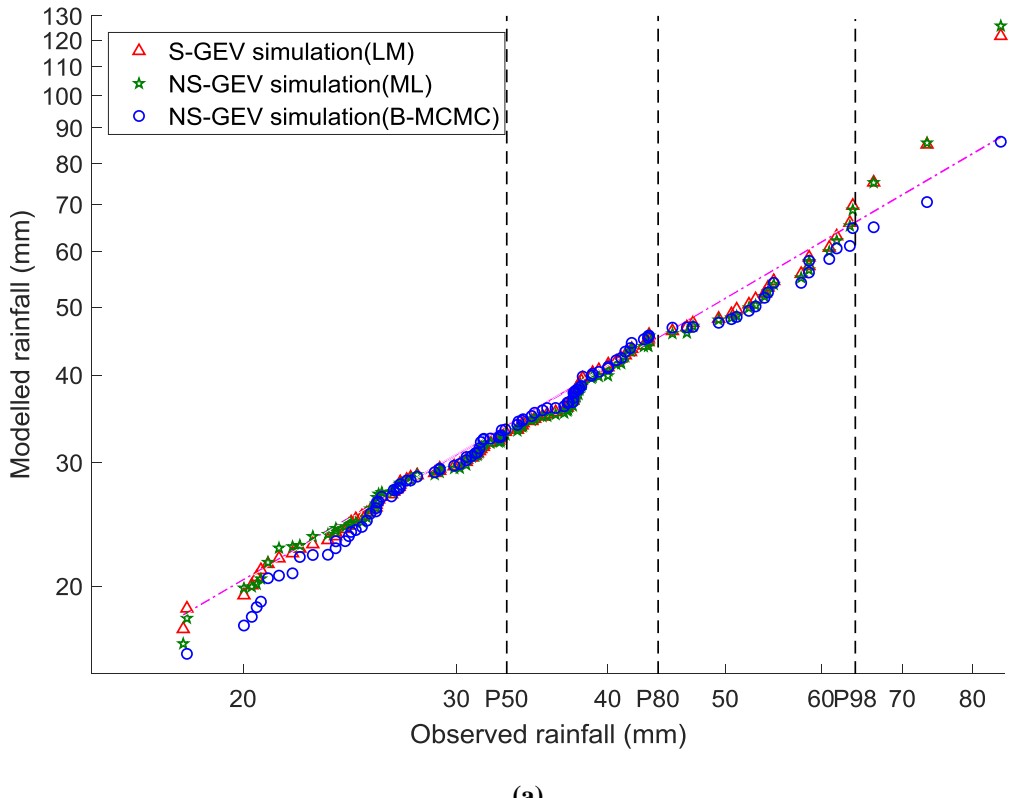

**(a)**





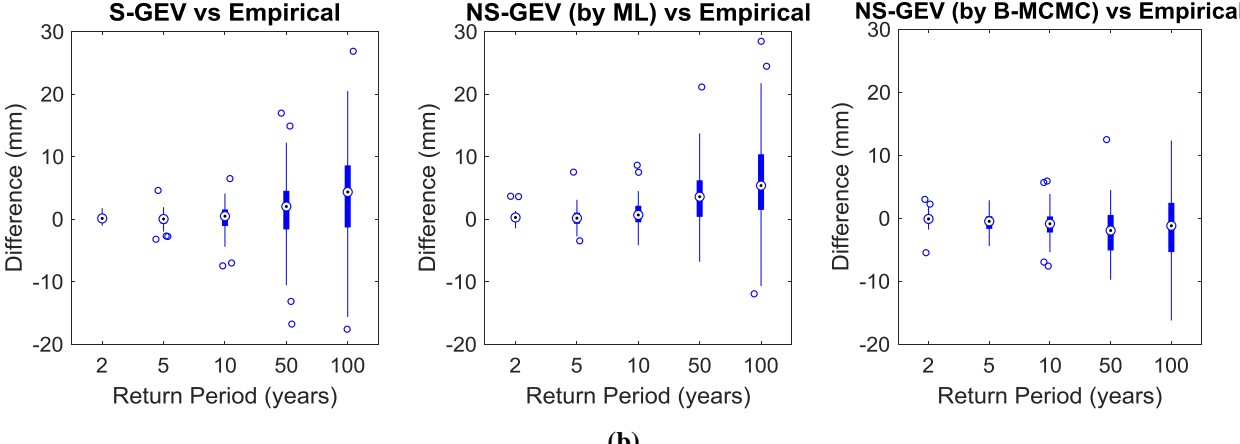

**(b)**

**Figure 8. Spatial distribution of the best fitting method for simulating AMDR's at different return levels: (a) L1; (b) L2; (c) L3 and (d) L4.**

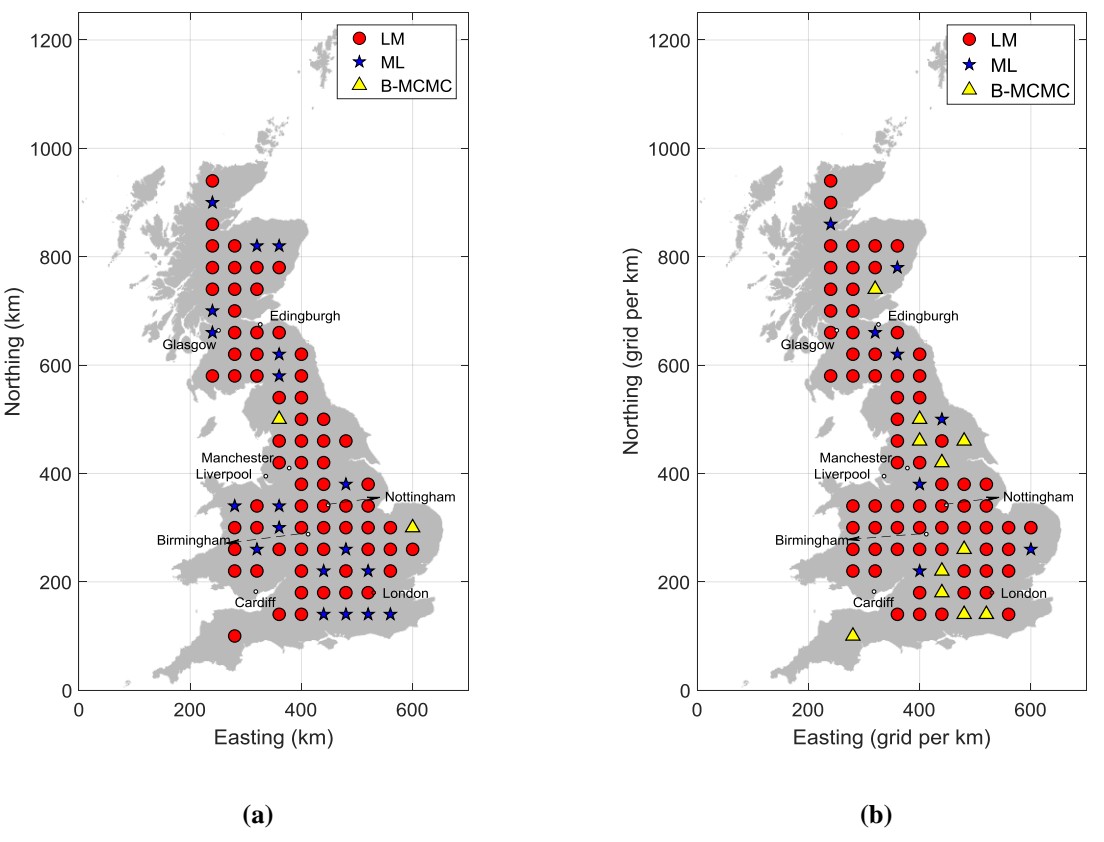

**(a)**                                       **(b)**





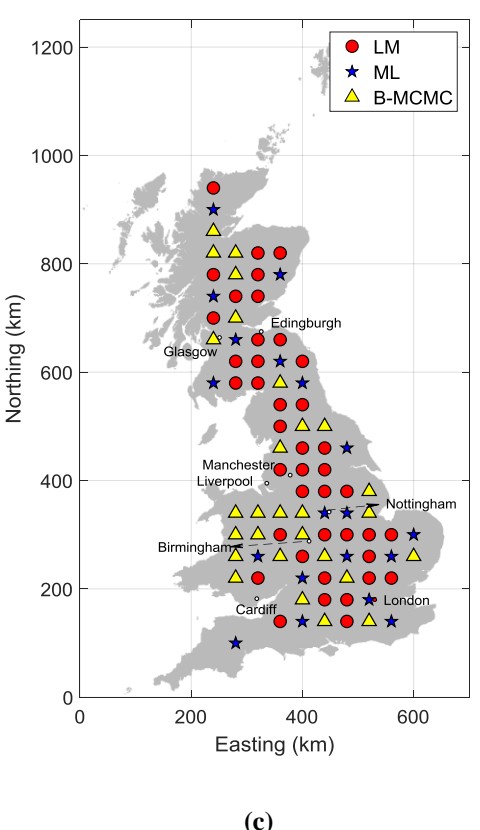

**(c)**

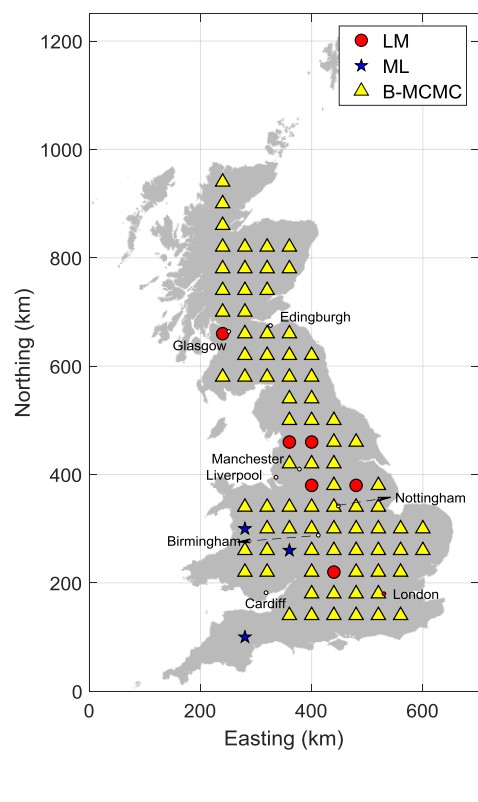

**(d)**