# Peer review of "Spatial Dependency in Nonstationary GEV Modelling of Extreme Precipitation over Great Britain"

_Hydrology and Earth System Sciences, 2020_

## Referee Comment (RC1) · Paolo De Luca (Referee) · 12 Apr 2020

**Review: "Spatial Dependency in Nonstationary GEV Modelling of Extreme Precipitation over Great Britain"**

**General comments:**

The authors provide a nation-wide study of historical precipitation by making use of stationary and non-stationary GEV models, with the aim to assess the spatial dependence of the different methods used. Although the study is interesting, it remains purely technical and therefore it is difficult to extract the impact-relevant messages and/or physical processes playing a role. Moreover, the study is also confined to a small geographical area. Performing a similar analysis for the European continent surely will help in improving the impact of the research. I therefore suggest rejection. You can address the above comments and/or submit the paper to a more technical-oriented journal.

Please see below some comments.

**Specific comments:**

Introduction is lacking a literature review on compound events related to hydroclimatic variables. This is needed as it will help in understanding the usefulness of your study. See below a few references to start with:

Zscheischler, J., Westra, S., van den Hurk, B.J.J.M. *et al.* Future climate risk from compound events. *Nature Clim Change* **8,** 469–477 (2018). https://doi-org.vu-nl.idm.oclc.org/10.1038/s41558-018-0156-3

De Luca *et al* 2017 Extreme multi-basin flooding linked with extra-tropical cyclones *Environ. Res. Lett.* 12 114009

De Luca, P., Messori, G., Wilby, R. L., Mazzoleni, M., and Di Baldassarre, G.: Concurrent wet and dry hydrological extremes at the global scale, Earth Syst. Dynam., 11, 251–266, https://doi.org/10.5194/esd-11-251-2020, 2020

I don't understand from the text if the data used are precipitation or rainfall.

L9 spatial dependency of GEV or extreme precip?

L20 engineering measures are usually needed to reduce socio-economic impacts, not to reduce impacts on natural processes.

L23-24 the second sentence of the introduction seems too much hasty and long. I suggest to 1) provide references of application of EVT to hydroclimatic data; 2) split the sentence in two; and 3) remove "flood" (and "etc.") from the list of hydroclimatic data, as this is a phenomenon that is eventually quantified by making use of data -> you can add "river flows" for instance.

L28 "(e.g. Gumbel, Fréchet, Weibull)" if you mention these three GEV distributions you need to give a bit more context about what they represent. Also change "e.g." with "i.e." since as far as I know they are the only GEV distributions available.

L32 sentence again too hasty. I suggest removing "e.g. precipitation, temperature, streamflow etc.," and explaining how hydroclimaitc variables are changing due to climate change. You can add 1-2 sentences with references.

L36-37 are you referring to nonstationary GEV? Scale and location parameters vary with time **and** other climate indices. Please amend.

L45 "depends on the elevation of study areas." Not clear

L52-53 "spatial dependency of the GEV distribution". Do you mean spatial dependency of extreme rainfall?

L53-58 too detailed for Introduction. I suggest shorten.

L69 "four related aspects of this study:" above you only mention 3 objectives.

L96 "The other two parameters can then be estimated by plugging back $\xi$ into Eq. (3)." Here 1) I cannot see Eq. 3; and 2) I suggest amending "plugging back" with something more appropriate for what is supposed to be a scientific study.

L100 equations need to be introduced in order.

L181 "we follow other researchers here". I suggest rephrasing.

**Technical corrections:**

L13 "are in favour of nonstationary GEV models"

L13 "as far as the annual maximum daily rainfall (AMDR) is concerned." AMDR is not a person, so please rephrase.

L20 "current engineering design storm practice". Not clear, please rephrase.

L155 please rephrase sentence.

---

## Author Comment (AC1) · 20 Apr 2020

We thank the referee De Luca (De Luca, 2020) for spending time reading the manuscript and providing valuable comments during the discussion that will certainly help improve the quality of paper via revision. While we agree and appreciate most of the points listed in the Specific Comments section, especially those regarding the referencing to 'compounded events', we strongly dispute the referee's view (in the General Comments Section) about the impact of the paper and the fitness to the journal. Our responses are as follows:

[Figure]

**1 Response to the General Comments**

Firstly, it is true that extending the research to Europe-wide or worldwide would have an impact in terms of revealing regional or global patterns and further encouraging exploring the climate dynamics behind them. It should be noted that, however, the methods we demonstrated are not the constraint that prevented us from doing so, it is rather the data availability that shaped the choice of using GB, a geographically smaller area but with sufficiently long data records (more than 100 years). This is in fact our overall idea behind this paper: we intend to devise, validate a novel approach and perspective for studying nonstationarity of GEV modelling associated with engineering designs, before applying it to wider areas where regional climate dynamics can be further studied, as the Spatial Random Sampling and nonstationary GEV modelling approach we provided in this paper are not limited by the geographical location but rather by the availability of the data. We agree that applying the methods presented to larger areas is equally important, and we are already exploring other data-reach areas, such as Australia, see Wang Xuan (2020). Another paper based on this conference contribution will be submitted shortly, which addresses the very comment made by the referee. However, given the scope of this manuscript, we don't think it is possible to include such case studies without diluting its main purpose.

Secondly, even the present paper focuses on Great Britain, the findings revealed are not only very relevant to the UK-based researchers and practitioners, they also demonstrate to researchers in other countries that how new features can be identified. The impact of this paper is mainly on the *quantification* of nonstationarity not only in the perspective of temporary changes but its *spatial* dependency, which contributes to improving the engineering designing approaches. As already shown in the paper, we found that, for the first time, not only do the most frequent events become more intensified, the extreme events also become more frequent in most part of the GB with refined spatial distribution. Such quantification will undoubtedly provide a basis to rethink how a new, more reliable engineering approach should be developed in view of climate change. And certainly, it will encourage readers/researchers in other regions to apply similar approaches related to the paper. In view of these, we find it hard to understand why the referee thinks there is lack of impact.

Thirdly, from our points above, the present paper fits the journal well based on our understanding as frequent readers of the journal. It may not be worth having yet another interesting discussion about the journal's scope, we can only point out that there are plenty of similar studies published in this journal, e.g. regional statistical analysis of hydrometeorological phenomena. Of course, this will be at the discretion of the journal editors.

**2  Response to Specific Comments and Technical Corrections**

We agree that discussion of 'compound events' is valuable and thus should be included in the introduction section. Although the main objective of the present paper remains as a univariate based nonstationary GEV modelling, readers should be informed well the relevance and importance of compound extremes, in particular, extreme flooding related to joint river-tide-storm surge impact. We appreciate the recommendation of the related paper which will be cited in the revised version.

We are thankful to the referee for other advices and suggestions of the necessary corrections. We will consider and implement them where appropriate in the next iteration.

**3  References:**

De Luca, P. (2020): comment on "Spatial Dependency in Nonstationary GEV Modelling of Extreme Precipitation over Great Britain" by Han Wang and Yunqing Xuan, Hydrol.

Earth Syst. Sci. Discuss., https://doi.org/10.5194/hess-2020-44-RC1

Wang, H. and Y. Xuan (2020): Temporal and Spatial Variation of Extreme Rainfall in Great Britain and Australia using the SRS-GDA toolbox, 6th IAHR Europe Congress, June 30th – July 2nd, 2020, Warsaw, Poland, accepted, https://doi.org/10.13140/RG.2.2.31765.27366

---

## Referee Comment (RC2) · Julien Worms (Referee) · 22 Apr 2020

**Referee report on manuscript HESS-2020-44 :**
**"Spatial Dependency in Nonstationary GEV Modelling**
**of Extreme Precipitation over Great Britain "**
**(by Han Wang & Yunqing Xuan)**

**Main comments**

The submitted paper proposes, in the context of flood frequency analysis, to assess that non–stationary GEV models are more suitable than stationary ones for modeling the extreme rainfall in the UK mainland territory. In my opinion, this is a relevant research question, that this paper tries to address.

Here are the problems I found while examining this manuscript. The main issue is the lack of clarity in the description of the methodology used.

1. An important issue with this manuscript is that *the authors do not provide a clear description of the type of data they are working with.*

   First of all, the reader has a hard work (I did) finding in the text the description of how the UK territory is separated in different zones, and the shape of these zones. Why considering only the mainland of UK ? Apparently, the territory is cut in non–rectangular zones , but *why not simply using rectangular zones, since only the very mainland part of UK is studied ?* (this is, in my opinion, an important issue with the paper). All the paragraph of the top of page 7 is rather obscure : what does the sentence "the focus is on the impact of location only" mean ? What do the lines 173–174 mean ? Do they mean that the zones have a common shape (that of Figure 1a ? Why this one ?) ? Why considering "randomized locations" ? It is not normal that such crucial description of what the study is about, is so badly described. And the reader can only wonder why a non–negligible part of the UK territory is not covered by the study (the seashores, and all the space between the different zones…).

   Second, the concept of "simulated samples" is not well described. If I understand well : (*i*) they start from real data from the GEAR dataset, and for each area, each model (stationary and non–stationary) and each estimation method, they produce estimates of their parameters (either $(\mu, \sigma, \xi)$ or $(\mu_0, \mu_1, \sigma_0, \sigma_1, \xi)$) ; (*ii*) with these estimates, they produce simulated datasets with these estimated parameters as inputs, one for each area/model/method ; (*iii*) they produce "indicators", which are the basis for selecting what they call the "best" fit for each area.

   If the simulation process is the one described above, why don't the authors present it in a clear manner in their manuscript ? This will enhance greatly its readability and accessibility.

   Finally, concerning this data description issue, in step (*ii*) above, I suppose that the authors produce *several* simulated samples (an not just a single one), but the text does not provide this precision… Only Figure 7b suggests that several simulated samples are generated (each one producing a quantile estimator for each return period). This example of lack of precision is symptomatic of the unclear way the methodology is presented in the paper.

2. Concerning the indicators which are used to compare the models (S or NS) or the estimation methods, I have some real concerns with them.

   Concerning the "Diff" indicator, its definition is unclear : line 153 of page 6, what do $y$ and $y'$ precisely refer to ? Examination of Figure 7b gives some hints of what the authors are doing, but Lines 151–153 of Page 6 are very mysterious and imprecise…

   The use of the Kolmogorov–Smirnov statistic in this paper is a real issue for me, particularly for the non–stationary modeling. The authors say, page 6, *"The test is carried out by comparing the empirical cumulative probability distribution with the GEV cumulative probability distribution"*. However, what does "*the* GEV" refer to ? If $X_1, \ldots, X_n$ ($n = 113$) denote one simulated sample for one area, for the non–stationary (NS–GEV) model, then the elements of this sample are precisely

constructed to not follow the same probability distribution ! So, if non–stationarity is strongly present, one cannot imagine that the "sample" $X_1, \ldots, X_n$ will fit a *single* GEV distribution. It is thus curious that the authors report (in lines 178–179) high $p$–values for the non–stationary model...

I also have some concerns with the stationary situation : in this case, it is true that testing that the sample is issued from a GEV distribution here makes sense. However it is very "surprising" that the $p$–values are all reported as being close to 1. *If* these $p$–values were correctly acquired and computed, and if most samples were correctly fitting a GEV distribution, then it is "common" statistical knowledge that the $p$–values should be uniformly distributed on the interval $[0, 1]$ (and if some samples were inconsistent with the GEV family, then the corresponding $p$ values would be *small*, not close to 1...). They don't here, according to the authors. My analysis is that the authors certainly make an unfortunately very common mistake, which is to compute $p$–values for the KS test when the target distribution was *estimated*, by using the KS test distribution when the target distribution $F_0$ is *known* and is not estimated from the data. This mistake makes the non–rejection (line 179, page 6) of the hypothesis "the AMDR follows the GEV distribution", not very reliable (note that, anyway, this sentence is strange : what does the article "the" mean ?).

Moreover, in line 147 of page 6, the authors do not precise (*i*) what sample is fitted to a GEV distribution ? (the simulated one(s) ? or the initial one from the GEAR dataset ?) ; (*ii*) on which sample is based the estimation of the parameters of the GEV distribution to which the sample (of (*i*)) is fitted ? This is an important point, which should probably explain why the $p$ values are found to be (artificially) so close to 1.

The authors might think that I have a rather severe judgement on their manuscript, however in my opinion it is normal to await a clear methodology description in a statistically based research work, if one wants the scientific findings to be accepted.

3. Another concern is that the reader has no idea of the influence of the choice of the region shape, or of the location of the centers; *how can one know whether the final findings of the paper will not (or will !) change if this common shape and/or choice of the location centers are changed ?*

4. Page 7 line 185, on which basis can the author say that the NS-GEV model "works better" than the S-GEV model ? (*i.e.* on which basis is the choice between the cross, and the squares, in Figure 3a, based ?). This is a crucial point of the paper .

How was the choice between $\xi > 0$, $\xi = 0$ and $\xi < 0$ made in Figure 3b ? On the basis of which sample(s) ? On the basis of a statistical test ? In particular, what situation can lead to the choice $\xi = 0$ ?

Figure 3d is interesting, however I do not understand why the values of $\mu_1$ and $\sigma_1$ (which are estimates) are multiples of 0.01 : have they been rounded ? If yes, why ? If no, please explain this curious situation...

Finally, in Figure 3c, what does $\sigma_1$ exactly refer to ? Does it refer to the estimate of $\sigma_1$ based on the initial GEAR data for the zone ?

5. In Figure 7b, how were the more extreme quantiles (return period of 50 or 100 years) estimated ? *And what do they mean / how are they defined in the non–stationary framework ??*

I will stop here for the general comments. If I admit being a non–specialist of the FFA topic, I nevertheless consider that the comments above are sufficient for not recommending publication of this manuscript in its present form, in HESS or elsewhere : many points need to be clarified.

**Minor comments**

- P1, lines 9–10 : sentence without verb.

- P1 , line 14 : what does "The most frequent AMDR as represented by the location parameter tend to be increasing..." mean ??

- P2, lines 49–50 : "In addition, .... nonstationary models" could be reconsidered, in regard to my general comment number 2 above...

- P2 , lines 63–67 : these sentences are not correctly written ("it then follows the introduction", "The specific focus on the spatial dependency of the methods tested the highlights of this study") or not very clear . Please rephrase them, since they are crucial for convincing the reader of the benefits of this work.

- P2, lines 31–45 : I am not sure the proposed bibliography of lines 31–45 is sufficient and/or sufficiently relevant, or should be completed.

- P3 , line 77 : the start of this subsection should be rephrased : in particular, the GEV cdf of equation (1) does not provide the "threshold value" $X_n$ as a function of the return level $F_n$, but the converse.

- P3 , line 85 : not sure the reference Nascimento et al is the most appropriate here... Prefer a well-known textbook.

- P3, line 86 : what do the authors mean by "independent parameters" ?

- P3 , line 87 : several problems (including "hence the name." : ??)

- P4 , line 89 : "involve" instead of "contain"

- P4 , line 91 : what does the "$r$-th random variable $X$" refer to ? An order statistics ?

- P4 , line 93 : $\beta_r$ is not an estimator, but the true value of the PW moment...

- P4, line 102 : this is absolutely not $Pr(X \leq x)$ , but the empirical cdf $F_n(x)$...

- P4 , line 104 : the notations $\sigma 1$ and $\mu 1$ are curious, why not using $\sigma_1$ and $\mu_1$ ? These are numbers, not vectors...

- P5 , line 110 : uppercase $X_t$, or lowercase $x_t$ ?

- P5, line 112 : "where $f(\cdot)$ is the univariate density function" is unclear . In the stationary setting, it is the presumed underlying GEV density. But in the non-stationary setting, it should be $f_t$, because the density depends on the value of the parameter $\theta = (\mu_0, \mu_1, \sigma_0, \sigma_1, \xi)$, and on the value of the year $t$ ! Can this "notation mistake" be considered as relevant of some sort of lack of comprehension of the backstage statistical theory by (one of) the authors ? Or lack of careful re-reading ?

- P5 , line 119 : the data is not described yet, so $n = 113$ is difficult to understand.

- Pages 5 and 6 , section on the B-MCMC method : this section is difficult to read, the notion of (random) step length is not defined, nor is the notion of "current state $S_t$". Either the authors should provide a preamble explanation of what a MCMC algorithm is run (and is about !), and be clearer in their explanations (for instance in an appendix), or they should provide an adequate reference for the statistical method that is run here. There is also some confusion between the state $t$ in line 121,p5, and the state $S_t$...

- P8 , lines 198–199 : please rephrase, unclear

- P9 , line 235 : this sentence is not clear. What does "dividing the AMDR values by their associated probability P into four levels" mean ?? Rephrase this part.

---

## Referee Comment (RC3) · Geoff Pegram (Referee) · 28 Apr 2020

Wang - hess-2020-44

Review:

This paper is about estimating extreme annual maximum rainfall throughout the United Kingdom, by comparing stationary and non-stationary extreme value distributions. The paper is interesting, if not ground-breaking, but is a possible contribution to the hydrometeorological armoury after major revision.

Regrettably, there are many irritating grammatical errors, which need correction. The

explanations are sometimes so poor that I had to re-read the paper 3 times before I could get the gist. The text introducing the mathematics is sparse and needs better explanation - you could have asked for help with the language from an English colleague to make it more readable. In particular, please be more informative in your figure captions, which are very terse and need textual expansion to help the reader follow what you have done, which is to present a puzzle difficult to solve. I nearly aborted at one stage. The paper is relatively short and some extra passages of explanation and referral to the text will aid the reader. For example, in the introduction and conclusion, it would help the readers who scan the paper before deciding to read it, if you tell them what S-GEV, NS-GEV, LM, ML & B-MCMC stand for. The captions in the figures are difficult to interpret and follow, and are as important as is the abstract to the potential reader. Also, throughout the paper, please include a space between paragraphs.

Nevertheless, I have been at pains to sanitise the errors I found and made suggestions as to layout, as in 'first aid'. My detailed remarks are itemised in an edited copy of the pdf, which I am returning with this review. I will append the major points from that version below my signature.

I recommend major revision.

Geoff Pegram

27 April 2020

Extended remarks follow – short ones can be found as captions in the attached pdf:

Line 77: 'For a given sampled area, the annual maxima series of the areal daily rainfall' . . . how do you determine the daily rainfall amounts from the 500 squares - arithmetic average? Also I suggest 'For a given sampled area of 500 km square, ..'

Line 109: 'The ML method (Myung, 2003) . . .' In my opinion, this is a trivial reference to the method of maximum likelihood, with which any reader of this article should be very familiar. If you do want a reference, go back to the originator Wilks - see be-

low. If you are serious about referencing, the idea originated from Gauss and Laplace! From Wikipedia: "Early users of maximum likelihood were Carl Friedrich Gauss, Pierre-Simon Laplace, Thorvald N. Thiele, and Francis Ysidro Edgeworth.[34][35] However, its widespread use rose between 1912 and 1922 when Ronald Fisher recommended, widely popularized, and carefully analyzed maximum-likelihood estimation (with fruitless attempts at proofs). Maximum-likelihood estimation finally transcended heuristic justification in a proof published by Samuel S. Wilks in 1938, now called Wilks' theorem." In the list of references, please substitute for Myung: Wilks, S. S. (1938). "The Large-Sample Distribution of the Likelihood Ratio for Testing Composite Hypotheses". Annals of Mathematical Statistics. 9: 60–62. doi:10.1214/aoms/1177732360

Line 155: ' boxplot is employed to indicate the under/overestimate the risk of extremes' Where is it?

Line 160: 'The Natural Neighbor interpolation method' please quote: Sibson, R. (1981). "A brief description of natural neighbor interpolation (Chapter 2)". In V. Barnett (ed.). Interpreting Multivariate Data. Chichester: John Wiley. pp. 21–36.

Line 163: please explain why Fig. 1a is a strange shape and discuss it in the text.

Lines 199 to 203: this passage is trivial and should be seriously pruned.

Line 216: 'About 56% samples are detected to have an increasing mu' I do not understand this statement - please enlarge - which figure?

Line 233: ' rainfall become rarer after 100 years, especially when sigma drops significantly' I do not see the reduction in sigma in figures 4, 5 & 6 - see my remarks on the figures

Line 370: In the revised version of this paper, please site the tables near the figures they refer to and also appropriately in the text.

In Table 1, you have the following headings: Nothing in column 1; Figure 6a – b – c in the rest In the row labelled sigma, you have '+2/100 yr; Unchanged; -2/100 yr'; what

does that mean? Sigma changes by +/- over 100 years? From what? As a ratio? The treatment of the parameters is very difficult to follow.

I have other remarks on Tables 1 & 2, not listed here – please see pdf.

Line 380: what are we to take away from Table 4?

Figure 1: 'Figure 1. The sampling area as shown in (a) the base data grids of a single sample; (b) the spatial distribution of the 88 samples.' I suggest: 'Figure 1. The sampling area as shown in (a) the base data grids of a single sample each containing 500 1 km squares of rainfall data; (b) the spatial distribution of the 88 sampling areas.'

Figure 3: Please explain here and to the reader of the text that mu1 and sigma1 should be defined as scaled parameters, relative to the central value of mu and sigma. This is VERY confusing and needs to be sorted out.

Please see my comments on Figure 4 – they need attention and I cannot reproduce them here usefully.

'Figure 8. Spatial distribution of the best fitting method for simulating AMDR's at different return levels: (a) L1; (b) L2; (c) L3 and (d) L4.' What are L1 to L4? The captions are all very terse and need to be expanded. Please explain in the caption as well as the text in Section 4.3. It seems that B_MCMC is far superior in fitting the rare events, as indicated in Fig. 7a, so make a comment.

Please also note the supplement to this comment:
https://www.hydrol-earth-syst-sci-discuss.net/hess-2020-44/hess-2020-44-RC3-supplement.pdf

[Figure]

**Supplement:**

[revised manuscript text omitted]

> In my opinion, this is a trivial reference to the method of maximum likelihood, which any reader of this article should be very familiar with. If you do want a reference, go back to the originator Wilks - see below. If you are serious about referencing, the idea originated from Gauss and Laplace!
> .
> From Wikipedia:
> "Early users of maximum likelihood were Carl Friedrich Gauss, Pierre-Simon Laplace, Thorvald N. Thiele, and Francis Ysidro Edgeworth.[34][35] However, its widespread use rose between 1912 and 1922 when Ronald Fisher recommended, widely popularized, and carefully analyzed maximum-likelihood estimation (with fruitless attempts at proofs).[36]
> Maximum-likelihood estimation finally transcended heuristic justification in a proof published by Samuel S. Wilks in 1938, now called Wilks' theorem."
> please substitute:
> .
> Wilks, S. S. (1938). "The Large-Sample Distribution of the Likelihood Ratio for Testing Composite Hypotheses". Annals of Mathematical Statistics. 9: 60–62. doi:10.1214/aoms/1177732360

10.1080/01621459.1949.10483310, 1949.

[revised manuscript text omitted]

*of NS-GEV?? here and in the summary*

[Figure]

(a)

(b)

(c)

Please explain here and to the reader of the text that mu1 and sigma1 should be defined as scaled parameters, relative to the central value of mu and sigma.

.

This is VERY confusing and needs to be sorted out.

Actual numbers of samples

|  | $\sigma_1>0$ | $\sigma_1=0$ | $\sigma_1<0$ |
|---|---|---|---|
| $\mu_1>0$ | 26 | 11 | 12 |
| $\mu_1=0$ | 3 | 28 | 1 |
| $\mu_1<0$ | 2 | 3 | 2 |

(d)

[Figure]

[Figure]

near Glasgow on Fig.3

**Figure 4. Observed and simulated AMDR of the sample located at 240km, 660km with stationary μ and σ.**

Easting or Northing?

[Figure]

what is the range of the grey simulation bands? why does it start small and end up large - even at the same AMDR?

[Figure]

near Bermingham on Fig.3

415

near Edinburgh on Fig.3

**Figure 5. Observed and simulated AMDR's of the samples located at (a) 360km, 660km with both increasing μ and σ.; (b) 400km, 340km with an increasing μ and an unchanged σ; (c) 280km, 220km with an increasing μ and a decreasing σ.**

near Cardiff on Fig.3

[Figure]

**(a)**

[Figure]

[Figure]

[Figure]

**(b)**

[Figure]

**(c)**

[Figure]

[Figure]

**Figure 6. Observed and simulated AMDR's of the samples located at(a) 400km, 620km with a decreasing $\mu$ and an increasing $\sigma$.; (b) 280km, 620km with a decreasing $\mu$ and an unchanged $\sigma$; (c) 280km, 510km with both decreasing $\mu$ and $\sigma$.**

**(a)**

**(b)**

[Figure]

[Figure]

, East of Edinburgh on Fig.3,

**(c)**

**Figure 7. (a) Q-Q plot of the simulated AMDR's of an example sample with the location index of (320km, 660km); (b) the difference between GEV-modelled and empirical daily extremes at different return periods.**

[Figure]

**(a)**

[Figure]

[Figure]

I suggest a zero level in each panel to help the eye

(b)

**Figure 8. Spatial distribution of the best fitting method for simulating AMDR's at different return levels: (a) L1; (b) L2; (c) L3 and (d) L4.**

what are L1 to L4?  please explain in the caption as well as the text in Section 4.3.  It seems that B_MCMC is far superior in fitting the rare events, as indicated in Fig. 7a

(a)    L1: P < $P_{50}$

(b)    etc.

[Figure]

[Figure]

**(c)**

[Figure]

**(d)**

---

## Author Comment (AC2) · 2 Jun 2020

We thank the referee Worms (Worms, 2020) for spending time reading the manuscript and providing valuable and in-detail comments that will certainly help improve the quality of paper via revision.

We particularly appreciate the points the referee highlighted regarding the use of the K-S test and the way of stating the return level in the context of nonstationary application – which will be further strengthened in the revision (see the detail response below). However, while we agree in principle with the referee that there are still spaces for clarification, we would argue that other comments, mainly related to 'sampling' and

'simulation', are unfortunately due to misunderstanding either of the context or of the technical approach employed. Our responses are as follows:

**1 Responses to the 'Main Comments'**

There are several terms used in this paper which have a slightly different meanings compared with the conventional cases where they appear. 'Sample/sampling' and 'simulation' are the two main terms which we believe are the major cause of the misunderstanding. We will rephase them to make it clearer thanks to the referee's feedback. Firstly, regarding the term "sample". We use "sample" to present the sampled areas which have the same shape and size and regularly distributed in the mainland of GB. These samples, or sampling areas, are generated by using a spatial random sampling toolbox which can randomize the characters (i.e., central location, size and shape) of the samples. We agree with the referee that ideally those areas should have been sampled in a pure random fashion with randomised location, shapes and sizes. In fact, the toolbox we developed and used can be easily set up to help achieve this. The reason we decided not to do it in this paper is based upon:

1. The scope of this paper is the non-stationarity in GEV applications in catchment hydrology. It does not intend to work fully fledged to reveal how the rainfall extremes vary over space and time continuously. This is also why a medium size is chosen to mimic a typical catchment. Certainly, there are other interesting features about the size, shape, local topography as well as the orientation of the shapes, which obviously deserves a separate study. In fact, we have already done such study which was briefly discussed in Wang Xuan (2020) with a full paper is due to submit. Again, in the present paper, the main interest is to investigate how non-stationarity vary with location and the setting (although not idea) will suffice for hydrological communities.

2. The choice of using 'sampled areas' instead of point or grid rainfall data is again for the consideration of hydrological applications. A no-space-left approach, as suggested by the referee, could have been easily implemented, i.e., just conduct and fit distribution grid-by-grid. This might be a better choice from a statistical analysis viewpoint but less so for hydrological applications.

Secondly, regarding the choice of GB and the GEAR dataset, we agree that the study area may not be as large as those studies aiming at revealing climatic variations at larger scale. Again, our choice is based on the practicality: 1) that the GEAR dataset is ideal for non-stationarity study as it contains long enough records with high spatial resolution, but unfortunately its coverage is limited, e.g. GB only; 2) that even within GB, the variation of GEV is remarkable as shown in the study and results are of great relevance for the scientific community in the UK; and 3) the methodology presented is not limited by the area or the dataset and can be extended to other areas with suitable datasets.

Thirdly, the argument around 'simulated samples' is where one of the main critical points is drawn from. We did not use the term "simulated samples" but we reckon the referee meant "the simulated AMDR". There are two scenarios with which 'simulation' is associated. The first scenario is that the once stationary GEVs are fitted to the samples of AMDR, the fitted distributions can be used to generate the 'simulated' AMDR from the inversion of the GEV's with input as the original empirical probability. This is a deterministic process. The 2nd scenario is when using the Bayesian MCMC method to fit non-stationary GEV's where multiple simulations can occur. These simulated AMDR's (the grey band in Figure 4,5,6) can also give an idea about the uncertainty of the nonstationary model. It should be noted that the AMDR's sampled from those regular areas do not involve any simulation (as previously explained). We regret that this has caused confusion and we will clarify this in the revised version.

Next, we appreciate the referee's remark on the use of the K-S test and the common problems may occur if it is not treated carefully. The present study follows largely the applications in many previous studies using this test for selecting GEV distribution in fitting hydro-climatic extreme datasets (Fischer et al., 2012; De Michele  Avanzi, 2018; Ayuketang  Joseph, 2016). And Many studies have shown that the GEV distribution fits well to extreme precipitation (Gong, 2013; Bonnin et al., 2006; Alila, 1999; Kysel'y  Picek, 2007). We did not include the detail sampling procedure (of candidate distributions/parameters using another type of MC simulation) for the calculation of the p-values as we were concerned about potential over-sized paper. This will be addressed fully in the revision alongside another test to support the choice of GEV.

For the introduction of 'diff', we proposed this as an additional measure to check the performance of the fitted non-stationary GEV's as it is very hard if not impossible to use the K-S test for the non-stationary models. The selection between S-GEV and NS-GEV or among GEV family is based on not only the p-values but also the "diff" measure. Figure 7b shows the difference of 88 sampling areas between modelled AMDR and empirical AMDR. Meanwhile QQ plot is also used for double checking by visual comparison (for example in Figure 7a). We will improve the definition and justification of the 'diff' measure, possibly with a less-confusing term in the revision.

Regarding the 'return level' in the context of nonstationary distribution, we thank the referee for highlighting this and agree that we should have been careful as to the definition of the return level. However, our intention is to make use of a concept of 'snapshots' looking at different exceedance probability calculated for the same rainfall value at both the beginning and the end (or any other middle point) of the period. This is a rather a Poor Man's approach, but it helps avoid the complex discussion of the return levels. This is also a common practice in the engineering community where practitioners tend to use reduced return levels (higher exceedance probability) to describe climate change impact. We will clarify this further in the revision.
Lastly, thanks for pointing out the strange look due to 'fixed' decimal places of Fig. 3d. This was inadvertently done, and we do apologise for the overlook as somehow the script we used truncated the values of the parameters to 2 decimal places. We will ensure the original values of the parameters be used in the revision version. And to Figure 3c, as we did not make use of any secondary process of GEAR dataset (do not have to), the estimation of parameters is all based on the original dataset.

**2  Response to the 'Minor comments'**

We are thankful to the referee for other advices and suggestions of the necessary corrections and will consider and implement them where appropriate in the next iteration.

**3  References**

Alila, Y., (1999) A hierarchical approach for the regionalization of precipitation annual maxima in Canada. J. Geophys. Res., 104(D24), 31645–31655.

Ayuketang Arreyndip, N. and Joseph, E. (2016). Generalized extreme value distribution models for the assessment of seasonal wind energy potential of Debuncha, Cameroon. Journal of Renewable Energy, 2016.

Bonnin, G M, Martin, D, Lin, B, Parzyok, T, Yekta, M, and Riley, D. (2006). Precipitation-Frequency Atlas of the United States, Volume 1, Version 4.0: Semiarid Southwest (Arizona, Southeast California, Nevada, New Mexico). NOAA Atlas 14.

De Michele, C. and Avanzi, F. (2018). Superstatistical distribution of daily precipitation extremes: A worldwide assessment. Scientific reports, 8(1), 1-11.

Fischer, T., Su, B., Luo, Y., and Scholten, T. (2012). Probability distribution of precipitation extremes for weather index–based insurance in the Zhujiang River Basin, South China. Journal of Hydrometeorology, 13(3), 1023-1037.

Gong, S. (2013). Extreme Analysis of an Annual Rainfall Dataset. In Advanced Materials Research (Vol. 610, pp. 2756-2760). Trans Tech Publications Ltd. Kysel′y, J, Picek, J., (2007). Regional growth curves and improved design values estimates of extreme precipitation events in the Czech Republic. Clim. Res., 33, 243–255.

Wang, H. and Y. Xuan (2020): Temporal and Spatial Variation of Extreme Rainfall in Great Britain and Australia using the SRS-GDA toolbox, 6th IAHR Europe Congress, June 30th – July 2nd, 2020, Warsaw, Poland, accepted, https://doi.org/10.13140/RG.2.2.31765.27366

Worms, J. (2020): Interactive comment on "Spatial Dependency in Nonstationary GEV Modelling of Extreme Precipitation over Great Britain" by Han Wang and Yunqing Xuan, Hydrol. Earth Syst. Sci. Discuss., https://doi.org/10.5194/hess-2020-44-RC2

---

## Author Comment (AC3) · 2 Jun 2020

We thank the referee Pegram (Pegram, 2020) for spending time reading the manuscript and providing valuable advices on both the technical approach and the editorial matters.

We are surprised and to be more precise, gratefully moved by the details into which the referee has pushed for when reviewing the manuscript – all being clearly manifested by the annotations made to the PDF. While we should not express personal feelings in an academic discussion, we certainly think such practice has become rarer in the current busy world and we are lucky to see there are still colleagues in the community

out there to follow this.

We accept the points made by the referee and we do like the new references recommended which we think will help consolidate the quality of the paper. We do apologise for the grammar and wording glitches in the present version and we thank for the candid comments made on them. We will ensure a thorough checking-all-through in the revised version.

In summary, all comments made by the referee alongside the corrections already indicated in the annotation will be dealt with following the advices of the referee.

**References**

Pegram, G. (2020): Interactive comment on "Spatial Dependency in Nonstationary GEV Modelling of Extreme Precipitation over Great Britain" by Han Wang and Yunqing Xuan, Hydrol. Earth Syst. Sci. Discuss., https://doi.org/10.5194/hess-2020-44-RC3